# Impact of an Enhanced Transtheoretical Model Intervention (ETMI) Workshop on the Attitudes and Beliefs Regarding Low Back Pain of Primary Care Physicians in the Israeli Navy

**DOI:** 10.3390/ijerph20064854

**Published:** 2023-03-09

**Authors:** Omri Besor, Ronen Brand, Ron Feldman, Yaniv Nudelman, Yair Shahar, Aharon S. Finestone, Noa Ben Ami

**Affiliations:** 1Israel Defense Forces, Medical Corps, Ramat Gan 5262000, Israel; 2Meuhedet Health Services, Haifa 3350127, Israel; 3Department of Physiotherapy, Faculty of Health Science, Ariel University, Ariel 40700, Israel; 4Maccabi Healthcare Services, Tel Aviv 6812509, Israel; 5Department of Physiotherapy, Tel Aviv Sourasky Medical Center, Tel Aviv 6423906, Israel; 6Department of Orthopaedic Surgery, Shamir Medical Centre (Assaf HaRofeh), Rhison Lezyon, Zerifin 7033001, Israel; 7Faculty of Medicine, Tel Aviv University, Tel Aviv 6997801, Israel

**Keywords:** enhanced transtheoretical model intervention (ETMI), low back pain, military, primary care physicians, return to physical activity

## Abstract

Low back pain (LBP) is a major cause of discomfort and disability. Physicians’ attitudes and beliefs influence the way patients with LBP are diagnosed and treated. The objective of the study is the assessment of military primary care physicians’ attitudes towards LBP and the effect of an enhanced transtheoretical model intervention (ETMI) workshop on them. We evaluated the impact of a 90-min ETMI workshop on the attitudes and beliefs of primary care physicians in the Israeli Navy on LBP. Outcomes were assessed using the Attitudes to Back Pain Scale in Musculoskeletal Practitioners questionnaire (ABS-mp). Participants responded before and after the workshop, and responses were compared to a control group of primary care physicians in the Air and Space Force. The intervention group included 22 participants and the control group included 18 participants. Both groups were heterogenic (gender, age, seniority). In both groups, primary care physicians reported the common use of non-steroidal anti-inflammatory drugs (NSAIDs) and over-the-counter (OTC) pain medications and often included physical activity and physiotherapy in the treatment plan. Physicians mentioned reassurance and suggestions of early return to physical activity as part of their appointment. There was a positive correlation between questionnaire items suggesting the physician tended to a biomedical approach and reporting the use of imaging modalities (r = 0.451, *p* = 0.005). After attending the workshop, physicians were significantly more likely to recommend an early return to physical activity (18 ± 0.48 vs. 16.4 ± 0.52, *p* = 0.04). An ETMI workshop had a minor impact on the attitudes and beliefs of primary care physicians regarding LBP, but a statistically significant impact was noted on return to physical activity recommendations. These findings may be important in the military setting.

## 1. Introduction

Low back pain (LBP) is a common phenomenon causing a heavy economic burden worldwide and affecting military environments in particular [1]. LBP has a variable course, characterized by recurrent painful episodes. In most LBP cases, no specific cause or clear nociceptive source is identified [2,3]. Furthermore, the association between structural spinal damage and the disability of chronic or non-specific LBP patients varies from nonexistent to weak [4,5]. Many LBP studies have found a strong and consistent association between psychosocial factors and chronic disability.

The two main LBP management approaches are the biomedical and the biopsychosocial models [6,7]. The biomedical approach is based on the hypothesis that pain is caused by tissue damage, therefore localization and treatment of tissue damage are essential. The biopsychosocial approach addresses biomedical concerns, in addition to the effect of other aspects including social connections, feelings of anxiety and depression, etc., on pain perception [1,8,9]. Consequently, clinicians’ attitudes regarding LBP may significantly influence treatment processes and outcomes [10].

LBP treatment guidelines recommend advising patients to stay physically active, provide positive reassurance and consider physiotherapy and complementary treatments. Additionally, it is recommended to avoid prolonged drug use (especially opioids and injectable steroids) and imaging (particularly in the first six weeks). Non-adherence to these guidelines comes at an economic, medical, and psychological cost that has been recognized by the World Health Organization [9]. Specifically in military settings, it also has an operational impact (days off duty) [9]. In the U.S. military, LBP was one of the most common causes of medical evacuation during operations in Iraq and Afghanistan [11]. Previous studies among military personnel demonstrated that short psycho-social education programs (Brief Psycho-social Education Program) reduce LBP-associated healthcare costs during a two-year follow-up period [11,12]. Enhanced Transtheoretical Model Intervention (ETMI) is an intervention that combines theory-oriented behavioral change counseling as well as dealing with barriers to performing desirable behaviors. Counseling is tailored to the patient’s level of readiness for change and aims to increase the patient’s productivity. This intervention has been proven to increase physical activity and reduce pain and disability among patients with chronic LBP [13,14]. The model is not part of the current medical training in the Israeli Defense Forces (IDF) and will be tested in this setting for the first time.

The attitudes and beliefs of primary care physicians towards LBP have not yet been explored in the IDF, and ETMI has yet to be attempted in this setting. This study examines the attitudes and beliefs of primary care physicians in the Israeli Navy and air force regarding LBP, and the impact of an ETMI workshop upon them. The research team hypothesized the physicians will hold a biomedical tendency, while after the workshop we expected a decrease in the prescription of medications, use of imaging modalities, and more recommendations for a quicker return to activity.

## 2. Methods

### 2.1. Study Design

This study used a quasi-experimental design. Primary care physicians, serving in the Israeli Navy and Israeli Air Force (IAF) were asked to participate in the study. Each participating physician received a description of the study and consented to participate. For the intervention group, the Navy physicians were invited to participate in an ETMI workshop in March 2021. The IAF physicians served as the control group and did not participate in similar workshops during the time of the study. Both groups completed the Attitudes to Back Pain Scale in Musculoskeletal Practitioners (ABS-mp) questionnaire to assess their sociodemographic characteristics, attitudes, and beliefs towards LBP and their clinical behavior when delivering care to chronic LBP patients. The questionnaire was distributed and answered online, anonymously, using Google Forms (Google Inc., Mountain view, CA, USA), before and three weeks after completing the workshop. Navy physicians who could not participate in the workshop were transferred to the post-workshop control group (8 physicians were transferred, 36%). Following the completion of data collection, an ETMI workshop was held for the control group.

### 2.2. Measures

#### 2.2.1. Attitudes and Beliefs towards LBP

The Hebrew version of the ABS-mp questionnaire was used to examine therapists’ attitudes and beliefs about back pain [10,15]. It consists of 19 items, using a 1–7 Likert scale divided into six subscales:
Limitations on sessions—Four items exploring practitioners’ policy towards limiting the number and length of patient–clinician encounters per episode of care (min–max = 4 to 28, where 28 = support unlimited sessions).Psychological—Four items measuring practitioners’ willingness to explore patients’ psychological issues (min–max = 4 to 28, where 28 = support psychological approaches).Connection to the healthcare system—Three items measuring attitudes towards the healthcare system and its available services and policies (min–max = 3 to 21, where 21 = feel connected).Confidence and concern—Two items measuring practitioners’ confidence in themselves and others regarding treatment and clinical limitations (min–max = 2 to 14, where 14 = confident).Re-activation—Three items exploring attitudes towards the return to work, daily activity, and increasing physical mobility (min–max = 3 to 21, where 21 = support re-activation).Biomedical—Three items concerning the belief that back pain has a structural cause as well as advice to restrict physical activity (min–max = 3 to 21, where 21 = support biomedical approach).

#### 2.2.2. Clinical Behavior

Four additional question clusters were included in the survey to assess physicians’ clinical preferences. They reported frequency of medication prescriptions, referrals to imaging, treatments provided during clinical encounters, and other treatment referrals. These questions were added based on a previously held study [16].

### 2.3. Intervention

A 90-min ETMI workshop was held with all participating Navy physicians (study group) to familiarize them with the method. ETMI focuses on the obstacles preventing patients from engaging in physical activity. Common obstacles are low motivation, fear of movement, and low self-efficacy. Physicians were instructed to include standardized sentences when addressing patient LBP. These sentences presented the importance of physical activity and its benefits in preventing LBP recurrence [17]. Physicians were encouraged to practice ETMI principles throughout the three weeks following the workshop.

### 2.4. Statistical Analysis

Descriptive statistics were used to examine the result distribution. Pearson/Spearman’s tests were used to examine the correlation among multiple variables with data distribution. A correlation between the ABS-mp’s subscale scores and clinical behavior was calculated from the combined data of both IDF branches due to similar characteristics (N = 37). Normally distributed continuous variables were compared using Student’s *t*-test for unpaired samples. The significance level was set at 5%. Principal component analysis (PCA) was used to investigate whether the clinical behavior items could be grouped under unique novel variables (factors). PCA enables for the grouping of a large number of variables to a specific common factor based on the explained total variance in the correlation matrix while retaining as much of the original data as possible [18]. The components identified were united and correlated with the ABS-mp questionnaire themes using Spearman’s correlation.

## 3. Results

### 3.1. Pre-Workshop

#### 3.1.1. Participants

The intervention and control groups included 22 and 18 participants, respectively, before the intervention. Both the primary care physicians’ groups were heterogenic (gender, age, seniority). Their characteristics are displayed in Table 1. There was a higher representation of female physicians in the control group. The groups were not significantly different in their pre-workshop attitudes and beliefs towards LBP (ABS-mp, Figure 1) and in their clinical behaviors. Detailed physician responses regarding their clinical behaviors and preferences before intervention are presented in Appendix A.

#### 3.1.2. Associations between Attitudes and Clinical Behavior

Physicians’ attitudes to set a predetermined number of treatment sessions in contrast to continuing the sessions despite no improvement (limited treatment) was positively correlated (r = 0.397, *p* = 0.011) with the physician’s connection to the healthcare system, and a sense of trust in other health providers inside their network (physician–employer relationship). The same sense of connection to the healthcare system negatively correlated with physicians’ confidence in the care they provide (r = −0.35, *p* = 0.027). Physicians with a tendency towards the biomedical approach were more likely to have lower confidence in the care that they provided (r = −0.348, *p* = 0.028). Physicians that reported a tendency to reassure their patients were less likely to limit the number of patient sessions (r = −0.543, *p* = 0.001) and had a lower tendency towards using a biomedical approach (r = −0.371, *p* = 0.024). Prescription of medications was positively correlated with physician biomedical approach tendency (r = 0.574, *p* < 0.01).

#### 3.1.3. Primary Component Analysis

Using PCA, the Kaiser–Mayer–Olkin (KMO) index of the analysis was 0.599, *p* < 0.001. Using the varimax rotation method, three factors were identified, reaching a total explained variance of 51.76%. Factor loadings are displayed in Table 2. Figure 2 includes Eigenvalues for components of each of the 14 clinical behavior questions.

The primary component (25.84% explained variance) was loaded by all the items assessing imaging and emergency department referrals. The second component (16.8% explained variance) consists of medications (benzodiazepines, cannabis, anti-depressants, and opioids), dry needling, and manual exam—loaded oppositely. The third component (10.69% explained variance) consists of OTC pain medication (paracetamol, ibuprofen, etc.), positively loaded, and reassuring the patient which is oppositely loaded.

Correlation analyses between the components and the ABS-mp’s sub-scores have demonstrated the following: the first component (imaging and ER referral) was positively correlated with physician’s trust in their health system and colleagues (r = 0.399, *p* = 0.014) as well as with biomedical tendency (r = 0.432, *p* = 0.008). The second component had no significant correlations with the ABS-mp’s sub-scales. The third component (OTC pain medication and inverse result of reassuring the patient) was positively correlated with the limitation of sessions (r = 0.361, *p* = 0.028).

### 3.2. Post-Workshop

There was a significant difference between the ETMI workshop attendees and the control group (*t* = 2.107, *p* = 0.042) in the return to physical activity score. The score was higher among workshop attendees, who were more likely to recommend a return to exercise and activity. The ABS-mp’s scores of physicians who attended the ETMI workshop and their controls are shown in Figure 3.

## 4. Discussion

This study investigated the attitudes and beliefs of primary care physicians in the Israeli Navy and the Air and Space Forces regarding chronic LBP and the impact of an ETMI workshop intervention upon physicians. Most enrolled physicians had under 5 years of clinical experience and were general practitioners without specialties. The two groups had similar characteristics. In Israel, the military physicians are usually graduates of a unique training program called “Tzameret”. In this program, every year 70 students start medical school at the Hebrew University in Jerusalem, before their military service (a reverse order compared to their high school classmates). During med school, they learn military-related topics such as trauma, but no special education regarding primary care or orthopedics besides the regular curriculum. After completion of an internship year following medical school (as required in Israel to complete their MD requirements) they are enlisted. The medical officers’ course is a training program in the IDF, performed combined for all physicians, and only then they will be divided into their units. Specialized medical courses for the Navy and Air Force’s unique environments are performed in-house and do not address subjects related to primary care medicine, already learned in medical school. Further medical training is also dependent on the needs of the specific unit where the physician is posted. It is important to note that most physicians are the primary medical figures in their units; second to them are the medics, and in some units, nurses and physiotherapists. Clinical decisions regarding the management of low back pain cases will be taken by primary care physicians and therefore they were chosen as our target population for intervention.

### 4.1. Physicians’ Attitudes and Clinical Behavior—Before ETMI Workshop

Many of the physicians reported that they often refer to physiotherapy and almost always examine their patients manually. Most physicians acknowledged that a large part of the treatment should include patient reassurance and encouragement to engage in physical activity.

The majority of physicians mentioned frequently prescribing pain medication to chronic LBP patients, in line with the American College of Physicians (ACP) guidelines that suggested a small improvement in pain and function compared to a placebo [19]. Other guidelines also encourage the use of NSAIDs for pain relief, even in chronic LBP [19,20]. It should be noted that a high biomedical tendency was shown to result in low adherence to guidelines among physiotherapists in Denmark [21]. We also believe that some of the medications prescribed are due to the patients’ expectations of physicians to be proactive and prescribe them pain relief medications and refer them to different imaging modalities, especially for chronic patients. These expectations might be influenced by attitudes and beliefs of primary care physicians towards LBP.

Imaging referrals (X-ray and CT) were mentioned by most of the physicians as being used sometimes or rarely (as detailed in Appendix A). These modalities were recommended for LBP if any red flags are suspected or in case the imaging is likely to change treatment course according to guidelines published in the European Spinal Journal in 2018 [20]. In a recent review, Traeger et al. mention that red flags may have a high false positive rate [22]. Overuse of imaging modalities may be attributed to the following three reasons. First, the physicians’ and patients’ belief that further imaging will localize the source of LBP [23]. Second, action bias; the physician wants to be proactive in treating the patient’s pain, and thus refer to different imaging modalities, despite not believing that they are necessary. Thirdly, the primary care physician’s insecurity in managing LBP leads to further testing for reassurance and stalling, during which the current episode of LBP will probably improve.

Most physicians (83.6%) reported a tendency to encourage patients to perform physical therapy, which is commonly recommended in several LBP guidelines [19,20,24]. Another consensus among clinicians was to reassure the patient and advise patients to return to exercise. Most guidelines recommend an early return to physical activity [19,20,24].

#### 4.1.1. ABS-mp Correlations

Pre-workshop, all physicians were asked regarding their personal interactions and treatment orientations using the previously validated ABS-mp questionnaire. The positive correlation between trust in the health system (the IDF’s Medical Corps and their colleagues) with their tendency to limit the number of sessions allocated for an LBP episode (treatment plan), might relate to confidence in other care providers such as physiotherapists or to clinician awareness of their own limitations in care. It is interesting to note that the physicians with higher levels of confidence in the care they provided had less confidence in the care of their colleagues and tended towards a less biomedical approach. This relation seems to suggest that physicians who are more knowledgeable of accepted LBP guidelines trust themselves more than others [25]. Similarly, physicians that were more likely to reassure their patients and acknowledge their concerns, were less likely to set a clear, limited treatment plan and were less oriented towards a biomedical approach. This information suggests the need to improve primary care physicians’ knowledge and self-confidence, while promoting a counselling system to support physicians and allow for easy and reliable referrals to other clinicians for treatment continuation.

#### 4.1.2. PCA

(1) Component 1—Imaging and ER

Through the PCA, an association emerged between physician referral to imaging (CT, Isotopic scan, MRI, X-ray) and EMG and referral to the ER. Both referrals were later positively correlated to the physician–employer relationship and biomedical approach of the physician. The more the physicians tended towards a biomedical approach, the more they pursued a structural or pathological cause of LBP. The timing of imaging and the specific modality chosen are important. This area requires further research and specific guidelines to assist physician decision making.

(2) Component 2—Medications, dry needling, and inverse manual exam

This component combined medication prescriptions such as benzodiazepines, opioids, cannabis, and antidepressants with performing dry needling and inversely with manual examination. We assume the combination of medications is less frequent in a primary care clinic, let alone in the military setting, and dry needling is performed only by trained physicians, and for a willing patient. Contrary to these rare treatments, the majority of primary care physicians reported performing a physical examination often. There were no clear associations of this component with other parts of the questionnaire.

(3) Component 3—OTC pain medications and inverse patient reassurance

Prescription of pain medications together with a negative score of reassuring the patient were part of the third component of the PCA performed. This component was later correlated with limitations on sessions. Our hypothesis consists of two parts. On the one hand, physicians that aim to shorten medical encounters with LBP patients tend to prescribe medications to terminate the session. On the other hand, physicians that have the time and patience to talk to the patients and address their concerns do not feel the need to limit the time for treatments, as these encounters might be longer than average and enable the patient to share his/her difficulties. Furthermore, the patient will need fewer physician encounters and less treatment time in the future due to the thorough initial encounter. This idea may have a considerable impact on appointment duration and frequency in the military. Due to the special relationship between patient and clinician in the military setting, there are grounds to study different physician approaches and their impact further (health and economic outcomes).

### 4.2. Physicians’ Attitudes and Clinical Behavior—Post-ETMI Workshop

We aimed to determine whether a short ETMI intervention would impact physicians’ attitudes and change their beliefs regarding LBP treatment. The results of this study demonstrate a significantly higher ABS-mp return to physical activity recommendation score by physicians who attended the ETMI workshop (18 ± 0.48) compared to the control (16.4 ± 0.52). Despite its significance, this difference is higher than the Standard Error of Measurement (SEM) for this subscale (0.994) but lower than its minimal detectable change (MDC—90%) value of 2.31. Therefore, this finding should be interpreted with caution. However, the SEM was calculated from a physiotherapist’s sample and these psychometric values may differ between occupations, as the mean and standard deviation of the physiotherapist ABS-mp’s reactivation score was 17.72 ± 2.5.

### 4.3. Limitations

The ETMI session held was short and the physician turnout was lower than expected, therefore drawing robust conclusions is problematic for this pilot intervention. Additionally, the initial control group had a higher percentage of female physicians, possibly due to a higher male percentage in the Navy to begin with. As with many questionnaire-based studies, there is a risk of response bias. For the PCA, the KMO and total explained variance are slightly less than the recommended values and larger samples are warranted in future studies [18]. The cohort included all the primary care physicians in the Israeli Navy and was matched by IAF physicians, yet the study was underpowered (58.76%). There is an extra drive to conduct the study on a larger cohort that may include both the caregivers and the patients.

## 5. Conclusions

This pilot study has shown that Israeli military physicians were less adherent to the latest guidelines in chronic LBP treatment. There was a greater tendency to be proactive, prescribe pain medication, and refer to imaging (biomedical approach) contrary to current LBP management guidelines. We have shown that physicians favoring a biomedical approach (as opposed to the biopsychosocial model) will refer to imaging and the ER more than their peers and have higher confidence in their employers and colleagues. Physicians in the intervention group that attended the ETMI workshop were more likely to urge their patients to be physically active. We believe that there is room for further workshops for IDF physicians, to teach ETMI and practice its use, alongside updating their knowledge of current LBP guidelines. Future studies with larger samples are warranted to validate the associations found between the attitudes and beliefs of physicians toward LBP with their reported clinical behavior.

## Figures and Tables

**Figure 1 ijerph-20-04854-f001:**
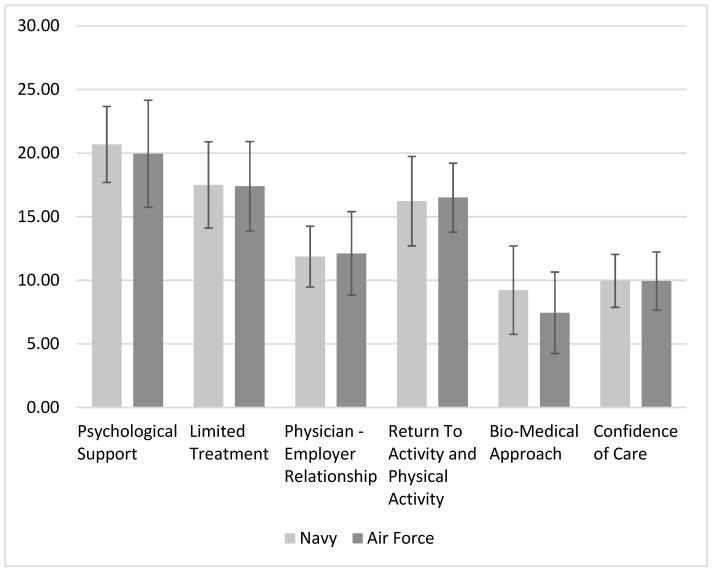
ABS-mp’s mean scores for every sub-scale before the ETMI workshop in both physicians’ groups. No significant differences were observed between groups.

**Figure 2 ijerph-20-04854-f002:**
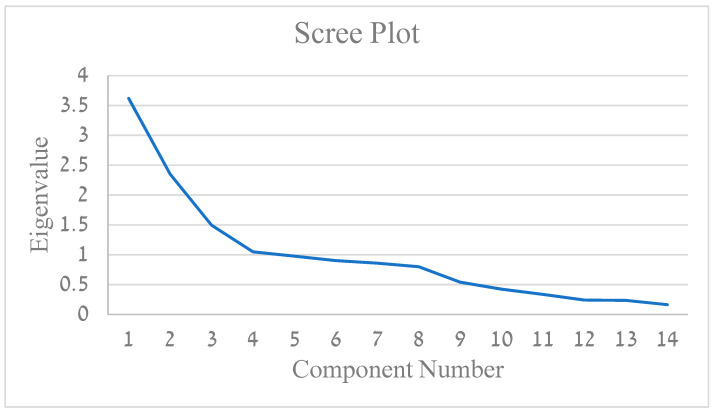
Eigenvalues for components for each of the 14 questions items.

**Figure 3 ijerph-20-04854-f003:**
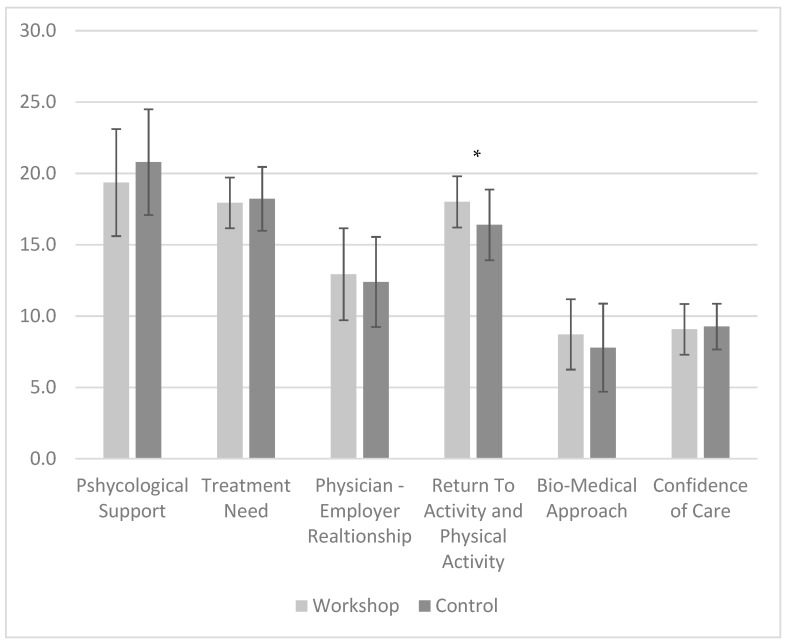
ABS-mp’s mean scores for every sub-scale following the ETMI workshop in both physicians’ groups. * Statistically significant difference (*t* = 2.107, *p* = 0.042).

**Table 1 ijerph-20-04854-t001:** Primary care physicians’ pre-workshop characteristics, by group.

		Intervention		Control		
		Number	Mean	Number	Mean	*p* Value
Age, Mean(SD)		22	32.64 (7.23)	18	31.65 (9.03)	*p* = 0.706
Gender	Male	18	81.80%	8	44.40%	*p* = 0.014
	Female	4	18.20%	10	55.60%	
Years of experience	0–5 Years	14	63.60%	16	88.90%	*p* = 0.186
	5–10 Years	4	18.20%	1	5.60%	
	Over 10 Years	4	18.20%	1	5.60%	

**Table 2 ijerph-20-04854-t002:** Pre-workshop factor loading and communalities for varimax rotated solution for 14 questions (N = 40).

	1	2	3
CT	0.83		
Isotopic Scan	0.79		
MRI	0.75		
X-Ray	0.67		
ER referral	0.66		
EMG	0.66		
Benzo		0.72	
Cannabis		0.69	
Anti-Depressants		0.64	
Dry Needling		0.64	
Opiates		0.60	
Manual Exam		−0.58	
Reassurance			−0.80
OTC Pain Medication			0.76

## Data Availability

Not applicable.

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
