# Peer review of "Impact of an Enhanced Transtheoretical Model Intervention (ETMI) Workshop on the Attitudes and Beliefs Regarding Low Back Pain of Primary Care Physicians in the Israeli Navy"

_ijerph, 2023, doi:10.3390/ijerph20064854_

Round 1
Reviewer 1 Report
This is relatively a small and simple study, assessing the approach of physicians to LBP and the effect of an educational program on their perception and decision-making process.
The small number of physicians and their different background should be taken into consideration. It is discussed in the limitations section, and the authors do mention the fact that it is underpowered.
Few remarks according to the article's sections:
Introduction
· Line 63 – consider rephrasing the sentence "… short psycho-social education program (Brief Psycho-social Education Pro- 62 gram) managed to reduce healthcare costs."
· Line 69 – it is worth mentioning that ETMI is not a part of the IDF Medical corps physicians' training, and therefore an explanation regarding the reason for choosing this specific group for additional training is needed.
· An additional sentence discussing the hypothesis of the study should be added.
Methods
· Line 77 - IAF – Israeli Air Force is the official name for this service, needs to be corrected.
· Line 80 - Please explain the prior experience and training of the physicians in relation to orthopedic conditions including LBP treatment. Additionally, a brief discussion regarding the composition of caregivers (medics, paramedics, nurses, physical therapists) will enable the readers to better understand the challenges of both caregivers as well as patients.
Results
· Table 1 – the number in the control group should be 18 instead of 17.
· It is worth adding some data regarding the type of units in which the relevant physicians are serving. Comparing operational or fighting unit's physician to a training unit might result in a different approach to the patients.
Discussion
· same as above regarding some elaboration needed regarding the caregivers' training.
· Line 257 – component 2 – this section needs clarification and language improvement.
· Line 264 – pain medications and not medication.
Overall, I think that this a simple study that has the potential to elaborate regarding the additional training needed for physicians with no orthopedic specialty. There are few corrections and editing, but overall it has a good potential.
Reviewer 2 Report
Dear Authors,
Good day,
I appreciated doing the review of the study. The text objectively and sustainably presents the problem, its antecedents, and the essential reference that led to the analysis. The results are organized concerning the objectives, and the discussion is relevant to the current object of research. They operationally demonstrate the impact of an ETMI workshop intervention upon physicians. They also present the study's limits and reinforce that changes in behavior and beliefs in the primary care of physicians can contribute to the multidisciplinary therapeutic approach and, consequently, to improving the conditions of people with LBP in the study group.
My only suggestion is to add a comment expressing that medical behavior and beliefs regarding the therapeutic process in LBP also reflect the patients' views and behaviors regarding what they expect from medical care, which is often restricted to drug therapy from identifying the possible cause investigated by images. I request this even if the study did not include the people assisted. I intend to indicate that the biomedical and biopsychosocial models are also reproduced by people who believe in and require specialized medical care supported by a specific care model. On the other hand, changes in the behavior of medical professionals are significant in influencing the coexistence of different care models needed in society.
This suggestion may be for future investigations comparing professionals and patients with LBP.
I hope I have contributed to the review of the study. Thank you for having been able to read the manuscript.
